# Commodification in Urban Planning: Exploring the Habitus of Practitioners in a Neoliberal Context

**Francisco Vergara-Perucich [1,*] and Martín Arias-Loyola [2]**

1   Núcleo Centro Producción del Espacio, Universidad de las Américas-Chile, Providencia 7500975, Chile
2   Departamento de Economía, Universidad Católica del Norte, Av. Angamos 0610, Antofagasta 1240000, Chile; marias@ucn.cl
*   Correspondence: jvergara@udla.cl

**Abstract:** The concept of habitus, as introduced by Pierre Bourdieu, serves as a lens to understand the subjective dispositions and perceptions that influence decision-making within the social realm. This study delves into the intricate relationship between urban planners' habitus and the commodification processes inherent in a neoliberal society. Through thematic analysis of semi-structured interviews with 27 Chilean urban planners, this research identifies typologies that capture their disciplinary stances on three pivotal urbanism facets: the city's conceptualization, the nuances of urban practice, and visions of utopia. A salient finding is the palpable tension urban planners experience, torn between the aspiration to foster a well-conceived city and the commodifying forces that shape decision-making. This commodification of the planner's ethos emerges as a byproduct of entrenched neoliberal institutional practices. This study delineates three distinct habitus typologies: the public, private, and academic urban planner, each exhibiting varied perspectives on the commodification of urban planning. Collectively, these insights shed light on the profound ways in which neoliberal paradigms influence urban planning, revealing both its disciplinary boundaries and inherent contradictions.

**Keywords:** habitus; theme analysis; neoliberalism; urban planning

## 1. Introduction

Neoliberalism exerts a significant influence on urban planning throughout Latin America, ushering in an era of market-centric urbanism and exacerbating socio-spatial inequality (Brites 2017). This phenomenon is notably evident in the case of Santiago, where neoliberal policies have played a pivotal role in shaping the trajectory of urban development (Fuentealba 2019). In Mexico City, the privatization of land and the implementation of market-oriented housing policies have been key drivers of urban sprawl (Salinas Arreortua and Pardo Montaño 2018). Similarly, in Sao Bernardo do Campo, Brazil, the ascendancy of the real estate sector has undermined progressive and redistributive planning efforts, highlighting the formidable challenges inherent in implementing inclusive urban planning within a neoliberal context (Souza et al. 2020). Collectively, these studies provide compelling evidence of the intricate and often contentious relationship between neoliberalism and the practice of urban planning in the Latin American context.

Chile, the cradle of global neoliberalism (Harvey 2005), initiated its urban model in 1979 with the National Urban Development Policy (Daher 1991; Gross 1991; Vergara-Perucich and Boano 2020a). This was later ratified by the regulatory framework provided by the 1980 Political Constitution, and consolidated through subsidiary urban development instruments. These shaped spatial transformations that, in cities like its capital Santiago, remain highly visible to this day. Specifically, the characteristics of the neoliberal urban model focus on the importance given to property and capital to guide urban development forms, assigning a dominant role to the market as a shaper of space. Meanwhile, the state assumes a subsidiary role for instances where the market does not participate due to

unprofitability (Anderson 2014; Castree 2006; López 2020; Pinson and Morel Journel 2016; Vergara-Perucich and Boano 2019).

It is due to this economic focus that analyses of neoliberalisation processes tend to centre on market, state, and consumer structures. However, they don't always focus on the actors implementing actions emanating from these neoliberal structures. In the specific case of urbanism in Chile, this approach is absent. Studies on the neoliberal city (Janoschka and Hidalgo 2014), the neoliberal urban model (Torres Jofré 2017), and its effects are abundant in literature. Yet, the subjective positioning of urban planners regarding disciplinary practice within the neoliberal framework remains open to empirical exploration. The complex and multifaceted relationship between neoliberalism and urban planning, as elucidated by scholars like David Harvey, is a subject of rigorous inquiry (Sager 2011). Neoliberal urban policies, driven by the mobility of investment capital, inter-city competition, and the ethos of public entrepreneurialism, pose formidable challenges to traditional paradigms of public planning. The neoliberalisation of social, economic, and political processes has ushered in a market-oriented orientation in urban development, planning, and governance, engendering inherent contradictions in the very governance structures of cities (Taşan-Kok 2012). Cities have emerged as central battlegrounds in the uneven and crisis-laden trajectory of neoliberal restructuring projects, with pre-existing institutional and spatial landscapes profoundly influencing the intricate interplay between neoliberal initiatives and the resultant urban transformations (Peck et al. 2009). The continuum of neoliberalism, ranging from radical system transformation to policy adjustments, is intricately linked to a broader typology of approaches concerning the reconfiguration of accumulation and regulation within advanced capitalist societies, with urban governance occupying a pivotal role in this evolving discourse (Jessop 2002).

This study focuses on this knowledge contribution, seeking to generate interpretations that inform typologies of urban planners' habitus to operate within a neoliberal disciplinary action field. Using Santiago de Chile as a case study is key, as its city-making approach still follows principles established during the neoliberalisation of the urban planning apparatus, primarily based on its dismantling and transfer to the market (Valencia and Marco 2007). Institutional design for a way of city-making is formed through experience and perceptions people have towards such experiences, forming a consciousness that guides attitudes and behaviours, similarly to studies in the sociology of art (Bourdieu 1990, 1997, 2014). This process and its persistence are maintained not only by institutional design, but by a mode of conduct of the professionals responsible for executing and designing cities, which could be termed habitus. In this sense, by formulating the habitus of a disciplinary body, predispositions and orientations for judgements can be explored, contributing to decision-making and relationships between actors (Yilmaz and Kuru 2021). Pierre Bourdieu defines habitus as systems of durable and transferable dispositions, whose structuring acts as generating and organising principles of practices and representations. These can produce outcomes without necessarily seeking such results consciously, but rather because of conduct inherent to an action field (Bourdieu 1991). Within Bourdieu's broad methodological contributions, habitus is inscribed within the theory of action, where agents acquire autonomy levels arising from adopting field rules, assuming a strategic disposition towards decisions (Silvina Daldi 2018). Thus, habitus is a social product of socio-historical origin that, having certain structures, operates as a social producer, enabling the continuity of certain agents' ways of doing in the social world (Rodrigo 2015).

The social field is understood as a local and social order upon which individuals position themselves to carry out games between actors, identifying their respective dispositions in them (Fligstein 2001; Stewart and Fielding 2021). Habitus synthesises individuals' approach to these games (Bourdieu 2004), allowing knowledge of what is taken for granted by these actors in each of the steps they take within the playing field (Reay 2015). In social research practice, habitus can be seen as an organisation technique of information emanated by agents, allowing the construction of a narrative about their preconceived and unconscious dispositions towards certain situations inherent to their tasks and daily life.

In this case, habitus is pertinent to characterising urban planners' subjective dispositions when operating within the neoliberal social world.

Since the social outbreak of October 2019, Chile has entered a deep political transformation process that could bring new ways of relating in society (Arias-Loyola 2021; Salazar 2020; Mayol 2019; Vassallo et al. 2021), starting with the redefinition of its social contract by discussing the formation of a new constitution. In this context, one of the key aspects is recognising city-making modes before this outbreak, to prospect potential changes in what could be post-neoliberal or welfare urbanism. In Chile's case, studies on urban planners' habitus do not exist, making this a key input for the times ahead.

Hence, the relevance of this article is justified, analysing under the habitus methodological framework the 27 semi-structured interviews conducted between January 2015 and August 2016 with various Chilean urban planners. This study explores their dispositions towards three fields related to their disciplinary tasks: a general view of Santiago de Chile, the practice of urbanism in such a city, and reflections on the role of utopia in urban thinking. These themes were triggers to identify specific views and predispositions towards these topics, seeking to identify and differentiate the habitus of these actors in the urbanism playing field. From these interviews, a coding and matrix organisation of results is carried out, analysed with the thematic analysis technique. The research behind this article is triggered by the following research questions: (1) How do the habitus of public, private, and academic urban planners in Chile manifest, and what are their defining characteristics in the context of neoliberal urbanism? (2) In what ways do these habitus influence the urban planning practices and decision-making processes, and how are they, in turn, shaped by the neoliberal urban context?

The objectives of this article are the following:

1. To explore and delineate the distinct habitus of Chilean urban planners within the context of a neoliberal urban landscape, focusing on three specific types: public, private, and academic urban planners.
2. To analyse how these habitus influence and are influenced by the practices, dispositions, and decision-making processes in urban planning and development.
3. To investigate the extent to which these habitus enable or constrain the realisation of urban visions, particularly in relation to utopian ideals, within the prevailing neoliberal framework.
4. To contribute theoretically to the understanding of urban planning practices in Chile by applying Bourdieu's concept of habitus, thereby offering a nuanced perspective on the interplay between professional dispositions and urban development dynamics.

From this analysis, a synthesis of results is constructed for each of the fields, and three typologies of the urban planner's habitus in Chile's neoliberal context are constructed. This allows the identification of three specific habitus: the public urban planner—based on technical knowledge of the norm but aware of its limitations, critical of the economic model; the private urban planner—critical of rent as the main objective of their practice but dependent on it, exercises their profession in contradiction; and the academic urban planner—with some creative and critical independence but without material results.

### 1.1. Chile: An Urban Society Commodified by Neoliberalism

Chile's long and narrow geography has several unique historical-political features that have made the country an unexpected key player on the global stage. Since the Spanish colonisation, the country's fate has been tied to the oligo-production and export of raw materials such as minerals, salmon, and forestry products. Currently, it holds the largest reserves and is the leading producer of copper, as well as being part of the "Lithium Triangle" alongside Bolivia and Argentina, boasting the largest reserves and global production (USGS 2020). This wealth has facilitated the socio-economic and territorial deepening of the country's extractivist development strategy, first implemented globally during the civic-military dictatorship that began in 1973 (Harvey 2005), and deepened during the representative democracy period from 1990 onwards (Arias-Loyola 2021; Barton 2002). For

decades, the world's first neoliberal experiment dazzled, and high macroeconomic growth rates combined with relative political stability led it to be considered "one of the greatest economic miracles of our time" by influential thinkers like Friedrich von Hayek (Ebenstein 2014, p. 300) and "an even more impressive political miracle" according to Milton Friedman (1994, p. 177).

Thus, for many international institutions, such as the World Bank (Foxley 2004), the extractivist neoliberalism model, colloquially known as "the Chilean model" was seen as a successful example of development based on the exploitation of natural resources, primarily minerals. Most of the mining exploitation occurs in the country's northern region, while the headquarters of multinational mining companies and the state-owned CODELCO are concentrated in Santiago, the nation's capital. Santiago captures much of the value produced in the extractive industries, as well as the exercise of the country's economic and political power. This has led to increasing inequality, where a large portion of the Chilean population and its territories live in chronic debt and job precarity due to the complete privatisation of social rights (education, health, pensions, housing, etc.) driven by the neoliberal model (Vergara-Perucich and Boano 2019, 2020b). The profound inequalities driven by neoliberalism, institutionally reified in the 1980 political constitution, have generated growing discontent among the Chilean population, especially in territories peripheral to political and economic power, such as Santiago's poorest communes and extractive regions, considered sacrifice zones.

This unrest finally erupted on 18 October 2019, initiating a series of protests in various cities across the country demanding structural changes (Arias-Loyola 2021), the specific consequences of which are still maturing. In the context of urban studies, Henri Lefebvre's conceptualization of urban life offers a critical and comprehensive perspective. Lefebvre, in his seminal work "The Production of Space" (Lefebvre 1974), posits that urban life is not merely a byproduct of spatial arrangements, but rather a dynamic process where space is continuously produced and reproduced through social practices. He emphasizes the centrality of everyday life in the urban experience, arguing that it is through the routine activities of its inhabitants that the city is both shaped and experienced. Lefebvre's approach transcends traditional notions of urbanism by integrating elements of social, political, and economic theory, thereby framing urban life as a multifaceted and dialectical phenomenon. This perspective is crucial in understanding the complex interplay between spatial structures and social processes in urban environments, highlighting how urban spaces are not only physical locales but also arenas of social action and interaction, deeply imbued with meanings and values (Lefebvre 1974). While the research for this article commenced prior to the social outbreak, the insights gleaned from the perspectives of its participants can illuminate the underlying causes of this historic Chilean event that began in October 2019. This period marked a significant moment when society collectively rebelled against an urban lifestyle that failed to align with their expectations, official narratives, and the stark contrast between the elite's optimism and the daily realities of the majority of citizens. We contend that the findings of this article offer valuable explanations for the social outbreak. What is clear is that the Chilean social outbreak, the pandemic, and the climate crises will produce changes in urbanism, where the naturalisation of the commodification of socio-spatial practices is under scrutiny. These changes will affect the ways cities are made, and it is relevant to situate the case with some historical context.

Urbanism in Chile emerged under the instruction of Austrian urban planner Karl Brunner in the 1930s, known as scientific urbanism (Bannen Lanata and Silva Pedraza 2016; Gurovich 1996), generating a virtuous urban production process for about 40 years until the 1973 coup, when the planning apparatus was suspended by the dictatorship. The regime focused efforts on benefiting business groups through various tax exemptions for construction, as well as reorganising cities by freeing up land from well-located areas occupied by settlers to create new urban peripheries in the nation's main cities, with particularly visible effects in Santiago de Chile, its capital city (Bohoslavsky et al. 2019; Celedon Forster 2019). The 1979 national urban development policy paved the way for an

urbanistic model centred on fundamentally mercantilist principles: (i) land is not scarce, (ii) housing scarcity will be solved by the market, (iii) all state intervention in cities should aim to improve land profitability, and (iv) the goal of urban development is to enhance the profitability of the real estate business (CNDU 2015; Daher 1991; Gross 1990, 1991; Vergara-Perucich 2019).

In this context, transformations in city praxis and the belief system for urban planners and architects' decision-making were also exposed to significant changes. These have been scarcely explored, but are directly related to the urban form of Chilean cities produced over the last 40 years (Vergara-Perucich and Boano 2020a). On the cusp of a change in the prevailing urban model, this study offers an historical record of how a group of urban planners practising their profession in Santiago de Chile evaluated their role in society. From these descriptions, it explores how their belief systems and decision-making principles could be categorised into habitus influenced by an ethos, based on the commodification of socio-spatial relationships.

*1.2. On Habitus, Lefebvre and Categories for Unpacking Urban Practices*

In the realm of urban studies, particularly when examining the lived experiences and practices of urban practitioners, Pierre Bourdieu's concept of habitus emerges as an invaluable analytical tool. Bourdieu's habitus, as Wagner and McLaughlin (2015) note, enriches the psychologically informed debates around social class, offering a nuanced understanding of how class structures influence individual behaviours and perceptions in urban settings. Lizardo (2004) further emphasizes the relevance of Bourdieu's framework in understanding the interplay of economic, social, and cultural capital in contemporary urban life. This approach is particularly pertinent in analysing how these forms of capital shape the practices and experiences of urban practitioners.

Warwick and Board (2017) highlight the utility of habitus in conceptualizing agency and the capacity to transform social structures. This aspect is crucial in understanding how urban practitioners navigate and potentially reshape the urban environment, influenced by their social and cultural backgrounds. Collet (2009) underscores Bourdieu's contribution to comprehending the complex processes of social interactions, central to the action learning that takes place within urban contexts. The habitus provides a lens through which the intricate dynamics of these interactions can be examined, revealing how urban practitioners engage with and are shaped by their urban milieu. Cottle (2022) argues that the greater the change in the social environment, the more salient the benefits of using habitus as a tool to analyse agents' behaviour. This is particularly relevant in urban settings, which are often sites of rapid and profound social change. By applying Bourdieu's habitus, researchers can gain deeper insights into how urban practitioners adapt to and influence these changing environments.

Wacquant (2012) advocates for combining an ethnographic methodological design with a Bourdieusian conceptual framework to enable powerful critical analysis and contextualized socio-political commentary. This approach is especially effective in urban studies, where ethnography can capture the lived experiences of urban practitioners, and Bourdieu's theoretical constructs can help interpret these experiences within broader social and political contexts.

Furthermore, Bourdieu's framework is instrumental in dissecting the impacts of neoliberalism on urban life. Neoliberalism, as a political project of state-crafting, emphasizes disciplinary workfare, neutralizing prisonfare, and the trope of individual responsibility, all serving the commodification process. In this context, habitus offers a critical lens to understand how urban practitioners navigate and respond to the neoliberal restructuring of urban spaces and policies. Incorporating Henri Lefebvre's triadic conceptualization of the city, practice, and utopia from "The Urban Revolution" alongside Pierre Bourdieu's habitus offers a nuanced methodological approach for studying urban life. While Bourdieu's habitus serves as an operative tool, grounding the analysis in the everyday practices and dispositions of urban practitioners, Lefebvre's concepts provide a broader conceptual

framework. The city, in Lefebvre's view, transcends its physicality, embodying a complex social construct. This perspective aligns with Bourdieu's understanding of social and cultural influences on individual behaviours, enriching the analysis of how urban environments are experienced and shaped. Lefebvre's emphasis on practice dovetails with Bourdieu's focus on habitual actions, highlighting the continuous production and reproduction of urban spaces through everyday activities. This intersection offers insights into the dynamic relationship between urban structures and the agency of inhabitants. Moreover, Lefebvre's notion of urban utopia, as an ideal driving urban development, complements Bourdieu's habitus by shedding light on how aspirations and visions of urban practitioners, influenced by their social and cultural capital, shape their interactions within the city. Utilizing Bourdieu's habitus for detailed, operative analysis and Lefebvre's concepts for a broader conceptual understanding, this combined approach provides a comprehensive methodological framework for examining the complexities of urban life, practices, and the pursuit of utopian visions in urban spaces.

Taking Lefebvre's contribution to urban studies, in order to articulate it with the use of the habitus to study practitioners approach to urbanisms in Chile, the definitions of city, practice, and utopia are operatively incorporated into the theoretical matrix of this research as follows:

City: Incorporating the insights of Henri Lefebvre and contemporary scholars, the city emerges as a multifaceted entity, transcending its physicality to embody a complex social construct shaped by historical, cultural, and economic forces. Lefebvre's portrayal of the city as a 'work of art' underscores its genesis from human activity and imagination, where social relations are not just represented. but actively constructed within its spatial confines (Purcell 2014). This dynamic, ever-evolving nature of the city, as a hub of diversity and a crucible for societal contradictions, positions it as a pivotal arena for revolutionary change and the creation of novel social spaces. Far from being a mere backdrop, the city actively participates in the drama of urban life, continuously moulded by and moulding the social practices of its inhabitants. This perspective aligns with Molano's assertion of the 'right to the city' as a potent response to neoliberal urbanism, empowering urban dwellers to reclaim and reshape their urban environments (Molano 2016). Thus, the city, in Lefebvre's framework, is not just a locus of human existence, but a living, breathing entity, intimately linked to the rhythms and experiences of those who inhabit its spaces.

Practice: In redefining the concept of "practice" through the lens of Henri Lefebvre's theoretical framework, it becomes evident that practice is a multifaceted and dynamic concept, central to understanding the interplay between individuals and their spatial and social environments. Lefebvre perceives practice not just as routine behavior, but as a critical component in the production and transformation of space, where space is simultaneously a result and a facilitator of these practices (Elden 2014). This perspective is integral to his spatial theory, highlighting the dialectical relationship between human agency and spatial structures. Practice, in Lefebvre's view, is laden with the potential to both replicate and alter social realities, thus playing a crucial role in the continuous evolution of urban and social landscapes. Coleman further elucidates this by characterizing Lefebvre's idea of practice as a critique of everyday life, suggesting that practice is an exploration of the human adventure, deeply embedded in the mundane yet significant activities that constitute daily existence (Coleman 2015). This approach elevates the concept of practice to a primary analytical tool for dissecting the intricate connections between space, society, and the minutiae of everyday life.

Utopia: In the realm of urbanism, Henri Lefebvre's concept of "Utopia" is intricately linked to his critical perspective on space as a socially constructed entity, highlighting the transformative potential of urban spaces in fostering revolutionary societal change. Lefebvre's interpretation of utopia diverges from the notion of a static, idealized model; instead, he envisions it as a dynamic, ongoing process that cultivates spaces conducive to the full realization of human potential and the establishment of a more equitable and participatory society (Coleman 2013). This approach directly challenges the norms of capitalist-driven

urban development, advocating for a profound re-envisioning of urban spaces as realms of lived experience where daily life is not merely accommodated but exalted. In this context, space becomes a deeply political entity, mirroring and influencing social dynamics. Pinder emphasizes this by noting Lefebvre's engagement with the "possible-impossible", a concept that emerged from a critical dialogue with other strands of utopian urbanism (Pinder 2015). Furthermore, Coleman elaborates on Lefebvre's multifaceted vision of utopia, describing it as 'dialectical utopianism', a blend of experimental and theoretical utopias (Coleman 2015). This dialectical approach underscores the notion of utopia as both a theoretical construct and a practical, experimental endeavour, continuously evolving in response to the complexities of urban life and societal development.

Using 'city', 'practice', and 'utopia' as analytical frameworks, the research conducts a review based on interviews with urban practitioners. This approach enables a deeper understanding of the ethos behind their decisions, their interpretations of the neoliberal context, and its influence on their practices. From these responses, the study categorizes the results and identifies various habitus.

## 2. Materials and Methods

This research is inductive, exploratory, and qualitative in nature, based on semi-structured interviews, using the case study research design proposed by Yin (2009). The semi-structured interview is a variant of the in-depth interview (May 2001). This method can be viewed as a purposeful conversation, where the interviewer and interviewee engage as equals in a horizontal dialogue on a specific topic (Seidman 2006). To effectively apply this method, the interviewer must be familiar with the areas they want the interviewees to address. One of the advantages of the semi-structured interview is that it allows for exploration of various perspectives on the topic at hand and provides new information or interpretations about the subject of study.

A semi-structured interview includes open-ended questions to encourage the development of responses without specific boundaries, although it is recommended to maintain certain time margins to facilitate the coding and interpretation of the answers. In this research, the purpose of using the semi-structured interview technique was to extract from the interviewee's various personal experiences as urban planners, to understand the conflicts and trends they observe in their daily practice of this discipline. The use of this method has been crucial to informing the analysis and subsequent construction of the urban planners' habitus under the neoliberal model.

The study has focused on professionals working in the Metropolitan Area of Santiago, considering that this urban space constitutes a complex urban system of 34 interdependent communes unlike any other in the country, concentrating about 40% of the national population and 60% of the GDP. Its complexity allows it to be seen as a case of a deeply neoliberalised metropolis, where urban practice is fluid and constant. Understanding the relationship between Santiago's urban planners and the neoliberal model sheds light on many of its contradictions, thereby facilitating its interpretation through Bourdieu's habitus method.

The sample was constructed following the judgment sampling technique explained by Marshall (1996), which involves creating an initial sample of interviewees aiming to interview those who will be most productive in answering the research objectives—in this case, to construct the urban planner's habitus in a neoliberal social field in the city of Santiago de Chile. Table 1 anonymously indicates a general characterization of the interviewed agents. Marshall suggests having a set of guiding questions that later allow classifying the answers by topics, but it was also accepted that the interviewees suggest topics or propose adding new interviewees. This is how three more interviewees were added to the sample. Since the aim was to gather information that informed a practice on the neoliberal field, it was important that the interviewees recognised neoliberalism as an existing condition in Chile, something that was not necessarily universal in the Chilean context.

Hence, a list of agents with a recognised analytical capacity in urban studies was needed, or at least with proven ability to be informed about recent disciplinary developments in Chile. The sample, therefore, consists of prominent authors of projects and research on the disciplinary field of urbanism, who have been part of renowned institutions related to urban development in academia, civil organizations, government, and the private sector. There was an imbalance between male and female respondents, which produces a gender representation problem, highlighting the relative masculinisation in the interpretations collected as a weakness of this work. No questions related to a gendered perspective were asked in these interviews, an aspect that is part of the methodological shortcomings of the research. In part, this problem arose because, from an initial tentative list of 40 interviewees, only 22 agreed to the interviews. Then, from that list of 22 confirmed interviews, 5 were added by suggestion of some interviewees in a snowball modality. By interview number 20, there was already a certain saturation of the sample, meaning the responses between different interviewees began to repeat in arguments and positions, ensuring that by number 27, the sample was already representative to develop a characterization by habitus.

**Table 1.** Sample of interviewees. Source: authors.

| Interview ID | Gender | Place of Interview | Realm of Practice |
| --- | --- | --- | --- |
| I1 | m | Santiago | Academia |
| I2 | m | Santiago | NGO |
| I3 | f | Santiago | Private |
| I4 | f | Santiago | Academia |
| I5 | f | Santiago | Public |
| I6 | m | Santiago | Public |
| I7 | m | Santiago | Academia |
| I8 | m | Santiago | Private |
| I9 | m | Santiago | Public |
| I10 | m | Santiago | Private |
| I11 | m | Valparaíso | Public |
| I12 | m | Viña del Mar | Public |
| I13 | m | Santiago | NGO |
| I14 | m | Santiago | Academia |
| I15 | m | Santiago | NGO |
| I16 | m | Santiago | NGO |
| I17 | m | Santiago | Public |
| I18 | m | Santiago | NGO |
| I19 | m | Santiago | NGO |
| I20 | m | Valdivia | Academia |
| I21 | m | Santiago | Private |
| I22 | m | Santiago | Private |
| I23 | m | Santiago | Private |
| I24 | m | Santiago | Private |
| I25 | m | Santiago | Private |
| I26 | m | Santiago | NGO |
| I27 | m | Santiago | Public |

The interviews were conducted in the cities of Santiago (24), Valparaíso (1), Viña del Mar (1), and Valdivia (1). From these interviews, narratives were collected on how urban planning operates in a neoliberal social world. Most of the interviewees' responses were recorded in audio, and all interviews included written notes. In five interviews, the recording device failed, which was only discovered when starting the transcription. The notes taken during the interview were crucial to address these issues. Questions were formulated to guide the interviews, involving a first part to understand the agents' relationship with urbanism and a second part with provocations on certain topics, attempting to foster a broad conversation on aspects such as politics, practice, utopia, and the capitalist city.

The limitations of this article may be understood in that, in 2019, Chile experienced a social outbreak, since when different interpretations of the causes have emerged in the literature. However, given that this study was conducted just before this outbreak, we believe that its contribution lies in offering potential causes that preceded the events of October 2019 in Santiago. Thus, this geographical bias is also an opportunity to provide new interpretive vectors for the causes of the social outbreak in Chile, from the urban space where it would originate three years after the completion of the fieldwork for this research.

During the development of the interviews, a preliminary analysis of the responses was carried out to extract some of the most common arguments about the relationship between urban planners and neoliberalism. After the interviews were completed, the notes were essential to constructing an initial narrative that was later populated with the content of the transcriptions. Finally, the resulting text was analysed using the Nvivo software, where codifications and content synthesis were applied, which also helped guide the thematic analysis process.

Thematic analysis is an analysis technique that involves organising findings or interpretations by general themes, then delving deeper by generating different thematic frameworks, usually applied to interview transcriptions or ethnographic processes (Braun and Clarke 2006; Mieles et al. 2012). The interviews were reviewed in the Nvivo software, selecting those responses with richer content, organised considering the three topics about the city, practice, and utopia, to configure the typologies of the habitus of the interviewed urban planners.

Habitus is constructed from individual subjective structures that accept certain levels of uncertainty in definitions. For this research, an operability of the habitus has been adopted based on the contributions of Luis Rodrigo (Benito and Miguel 2012; Rodrigo 2015), who states that habitus are theoretical constructions in the form of typologies composed of empirical data and, thus, constitute ideal types that seek to highlight common characteristics that could be found in other agents of reality. Purity as a possibility of results is rejected, since its composition is based on subjective dimensions associated with the agents' perception and objective dimensions based on their experience (Rodrigo 2015). In this case, urbanistic habitus will be understood as the subjective dispositions of urban planners to operate and relate within a neoliberal social order.

To construct the typologies of the habitus (Figure 1), some agents' positions on certain topics for specific fields are presented below. Following the methodological strategy of thematic analysis, in the first order, it is allowed for the interviewees' voice to construct the narrative, and then the typologies that emerge from the analysis of the sources are configured.

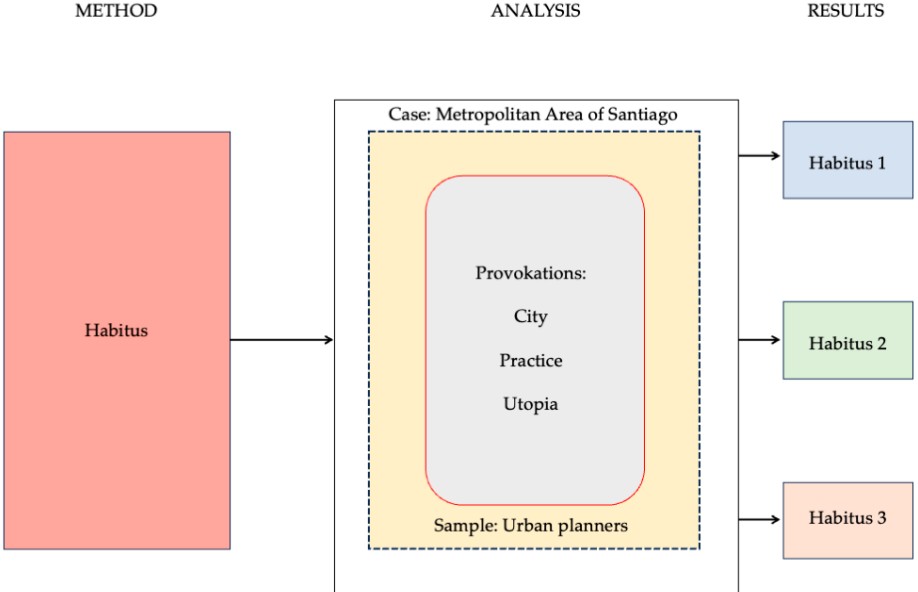

**Figure 1.** Diagram of the methodology of this research.

### 3. Results: Fields of Action for Urban Planners under the Chilean Neoliberal Urban Model

After analysing nearly 30 h of audio and synthesizing the interviews into 20,000 words that best addressed the objectives of this research, three fields were configured based on the interviewees' experiences. These fields later converged into an approximation of an urban planner's habitus in a neoliberal context for different typologies. It is anticipated that there has been a certain level of convergence in most of the actors' interpretations of the proposed fields. However, the divergences found allow for a more nuanced understanding of the habitus, which will later be used to construct the typologies. As part of the coding process, it was found that the sentiments when discussing these topics were mostly negative, meaning the emotional charge of the phrases and arguments stemmed from feelings of powerlessness, pessimism, and even anger in some cases (Figure 2).

A second initial approach to the interviews from the coding process allowed for the identification of the most frequently used concepts during the interviews. The word cloud in Figure 3 indicates some of these repeated expressions, which help identify certain ways of presenting ideas during the interviews. On one hand, the word "state" stands out, where most of its use describes a negative connotation, often alluding to an unsatisfactory role towards the city under a pro-business bias, which aligns with general definitions of a neoliberal state (Harvey 2005). There seems to be greater interest in the concept of "land" as a fundamental element of the neoliberalisation process, as it appears that for the interviewees there is little understanding of the implications of land for the development of a just society.

Then, a third frequently used term was "example", associated with trying to demonstrate opinions with cases. This may reflect a need to show that some reflections correspond to urban reality, which was also seen in the less grounded, more imaginative reflection, detached from the strictly empirical. These initial insights are fundamental to constituting the different habitus that will later be explained.

Next, the dispositions that illustrate the agents' positions for the different proposed topics are presented.

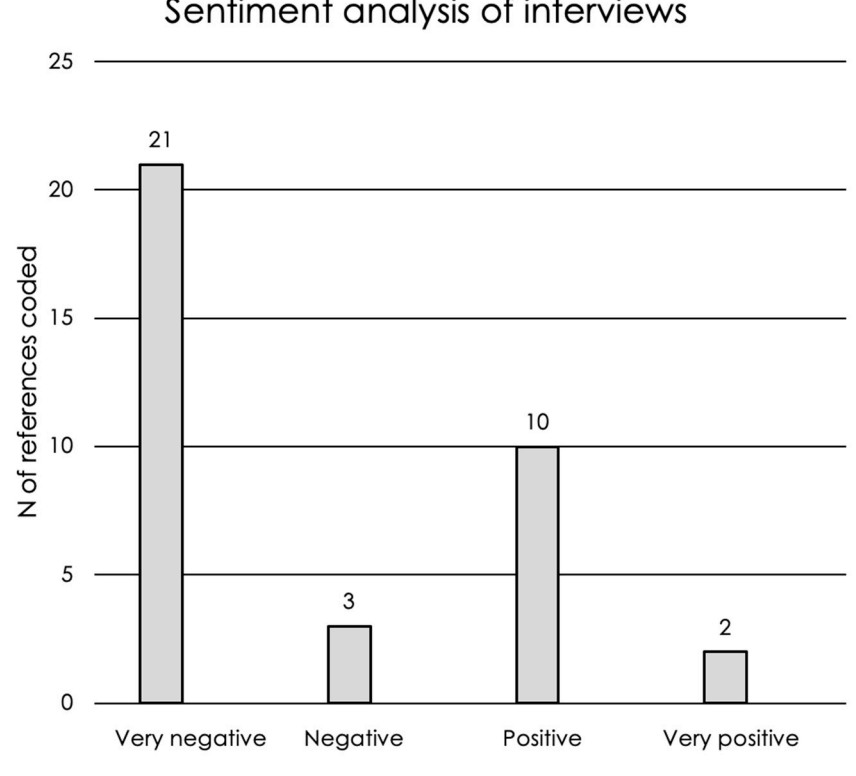

**Figure 2.** Classification of text references in relation to feelings expressed during the interviews.

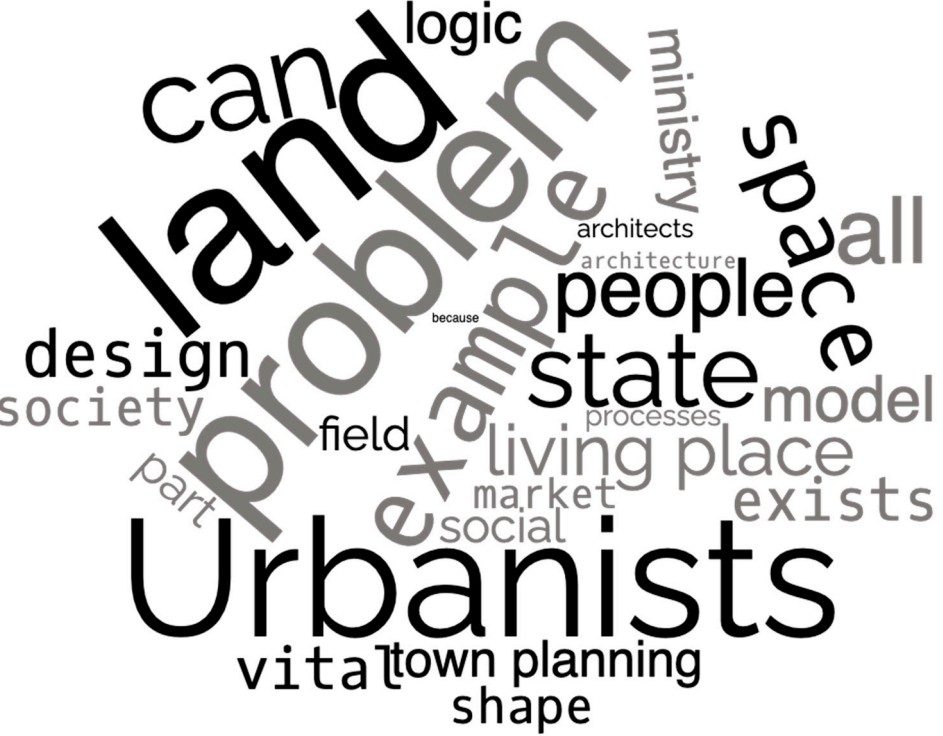

**Figure 3.** Most used words by the interviewees. The words city, urban, Chile, and Santiago have been excluded from this list as they are redundant being part of the questions.

### 3.1. Santiago de Chile as an Object of Study

Most of the interviewees live and work in Santiago de Chile. Their perception of the city tends to be of an urban area with profound problems stemming from its spatial configuration, leaning towards inequality and class division of urban space, which results in a certain quality in some places and a very different one in others:

> "[Santiago] is the capital of inequality. There are many worlds where all the urban evils and attributes can be found. It's also a ticking time bomb, with no decompression controls. But it's an example of why the market can't take care of the city because it produces bubbles of good and bad quality of life" (I11).

> "Santiago is a diverse or rather heterogeneous city. It's an administrative centre comparable to European cities in the eastern sector, very good in terms of quality. But the other three-quarters of the city are very indivisible, where the majority of people live; it's a divided city where, due to the concentration of activities and distances, the poor and the rich don't interact" (I4).

Regarding Santiago, the emphasis tends to be on its structural inequality, as a result of an economic model that is fundamentally unequal and then represented in built space. While the view of this configuration tends to be relatively negative, there are also some more optimistic views that do not fail to recognize inequality as a problem, as indicated by interviewee number 12:

> "Santiago is a candidate to become the capital of Latin America. Santiago's economic process is very dynamic and is improving. It has a bad reputation, though, and it's worth getting rid of that bad reputation. For example, the accumulation of wealth is astronomically different from other regional cities, and that clearly explains the social segregation it has" (I12).

In line with this statement, there's the recognition of the city as an entity produced from a neoliberal model that turns all public assets into commodities for sale for the profit of economic groups:

> "Santiago had to be this way, under capitalist logics. Not because it doesn't aspire to this, but because the profits the city provides are a means for the financial system to also obtain profits. Then, the subsidiary logic of the State perpetuates these forms of land profitability. The bad thing is that the profits go into the pockets of a few, there's no distribution or redistribution of wealth. With this, the marginality of citizens appears, and in short, society as a whole is speculated upon... The value of the land in polarization and capitalization is relevant to most, but it seems that many believe this phenomenon is very indirect. It sounds abstract to most. If it doesn't work with more systemic awareness, it's difficult. Only when these problems are linked to people's social problems and the information is simplified do people understand the magnitude of the problem" (I7).

The statements of interviewee I7 indicate a lack of knowledge about the causes that gave rise to this city, something that is not only attributed to the citizens but also to decision-makers. "The city is the result of economic and political social relations; it's a reflection of a market development model that is not unjust per se, but we, the actors behind the market, are unjust" (I6). In general, it is observed that the interviewees recognize certain causes that produce an unequal urban form. However, there is little self-criticism, and the diagnoses tend to blame others who did not make the right decisions when, in practice, the interviewees are opinion leaders and often have also been decision-makers.

Regarding how the city became so unequal, there are different convergent approaches towards an explanation that goes beyond the economic model. There's the governance issue, "Santiago is 32 estates, an unregulated metropolitan plot" (I10). This refers to the national municipal logic, where each commune must self-finance its management, which is another way to understand inequality since communes that concentrate high-income households or jobs have an easier time gathering resources than other communes. To this

municipal governance problem is added the absence of a coordinating authority for such an extended metropolis, "it's more than proven: social segregation is harder to manage the larger the city is. It's evidence that needs to be addressed because it impacts everything" (I9). Thus, there's a view that the city's expansion growth contributes to making it difficult to develop solutions that solve the problem of spatial inequality, in which political decisions have been fundamental:

> "It has been the lack of a multidisciplinary vision that has made Santiago what it is today. In Chile, development has been led by the Ministry of Public Works and the Ministry of Finance, which is thinking about the city macroeconomically, rather than a holistic vision. For example, the public works carried out seek productivity and not quality of life" (i16).

Santiago, perceived as a city produced for the interests of capital and its growth, was a common theme during the interviews. While it is not entirely declared, the interviewees tend to find causes of the urban problem in another group of actors: those who work in the state blame the market; those in professional practice blame the state; while academics blame both. Certainly, in academia, there is a more complex diagnosis, but from the experiences of the state and the market, it can be seen that there is also a latent reflection on the underlying problems. Despite sharing the disciplinary field of urbanism, there seems to be little dialogue between the diagnoses which, beyond the blame, tend to converge: an unequal, deregulated city in crisis with complex solutions, which will necessarily demand cultural changes, as indicated by interviewee I4 and interviewee I17:

> "In Santiago, the relationship between the city and equity is hard to achieve. It seems that Chilean society is not very willing to integrate: the poor with the poor don't want to integrate, and the same goes for the rich, obviously. People don't want to mix because of a lot of prejudice about others. This is a cultural problem; there's a real psychosis about the fear of the poor, which is accompanied by an invisibility of the other" (I4).

> "It's a sick living being. It has functional interdependence, despite segregation, which affects even moods. The city is a whole; they are like states of consciousness typical of a city that lives in social injustice, like an organism" (I17).

Regarding the city of Santiago, even the most optimistic views emphasize the importance of resolving the structural inequality that affects it and spatial segregation. However, there is a saturation of arguments, meaning the descriptions of the city are repeated with different emphases but in the same direction, indicating that most of the interviewees tend to get stuck in the neoliberal economic model as the cause, and then it was difficult to achieve deeper insights. Even the agents participating in professional practice pointed to neoliberalism as the main cause of Santiago's form.

*3.2. Field of Urban Practice*

The urban practice in the Metropolitan Area of Santiago presents a compelling case study in urban governance, characterized by an intricate interplay of political, economic, and professional dynamics. Politically, the region is governed through a multi-tiered system involving national, regional, and municipal authorities, each wielding varying degrees of influence over urban planning and policy decisions. Economically, the area has been significantly shaped by neoliberal policies since the late 20th century, with a strong emphasis on market-driven development and private sector involvement in urban projects. This economic approach has led to rapid urban expansion and significant private investment in the region, but also to challenges such as spatial segregation and housing affordability issues. Professionally, urban planning in Santiago is a field dominated by a mix of government planners and private sector professionals, with the latter often playing a pivotal role due to the market-oriented nature of urban development. However, this professional landscape is not without its tensions, as planners frequently navigate the complex terrain of political interests, economic pressures, and social equity concerns.

The resultant urban form of Santiago thus emerges as a product of these multifaceted and sometimes conflicting influences, reflecting broader trends in urban governance and development in the global south.

Urban practice can be reviewed mainly from research associated with academia, the development of projects associated with the professional field and civil organizations, and the management of public resources associated with state practice. Regarding the state, there is some internal confusion due to the excessive number of entities involved in the city without clear coordination, where "there is no established model, only certain guiding guidelines are followed" (I2). Then, practice and decisions depend on the specific requests of the capital provider: "methodologies depend on the client's requirements. The programmatic requirements are sought to be grounded" (I3). The adaptability of these methodologies is expected for commissioned professionals, but there is a question of whether there is real decision-making power over the results or if there are only creative spaces and real decision-making in areas that do not touch profitability, both in the private and public sectors:

> "To request a budget [from the Ministry of] Finance for urban projects, the Ministry of Social Development, through the National Investment System, favourably recommends an urban project or initiative, always depending on social profitability. This is an economist view, and this view deepens inequality for a simple reason: it concentrates investment in cities with larger populations. There is no evaluation of socioeconomic conditions, and this generates segregation, leading to projects that were good but end up being bad from an urban point of view. Then, due to fragmentation, the Ministry of Social Development has few resources to evaluate an urban project, but they also have a lot of needs and tend to use the minimum in terms of urban standards. This criterion obeys another economist reason, at a lower cost, higher social profitability, and in the end, that defines the development of an urban project" (I11).

The budgeting process of Santiago's communes, deeply rooted in a self-managed funding system, plays a crucial role in perpetuating urban inequality within the metropolitan area. This system, primarily reliant on local property taxes, inherently favours wealthier communes with higher property values, enabling them to generate substantial revenue for improved public services and infrastructure. In stark contrast, less affluent communes, constrained by lower property values, struggle to raise adequate funds, leading to poorer services and a consequent reinforcement of socio-economic disparities. Although central government transfers provide some financial relief, they often fall short of significantly bridging the gap between rich and poor communes. The budgeting process itself, a blend of technical assessments by professional municipal staff and final decision-making by elected officials, is influenced by a complex interplay of professional expertise and political dynamics. This scenario highlights a critical challenge in urban governance: addressing entrenched inequalities amidst a decentralized funding structure that intrinsically links a commune's financial health to its property market, thereby exacerbating the divide between the affluent and less prosperous areas of Santiago.

Another aspect mentioned is the problem generated by the fragmentation of the entities that operate on the city, especially when making specific diagnoses to improve project outcomes. This also occurs at the level of disciplinary fragments, shaped in a university education model that does not promote interdisciplinarity, nor does it promote the integration of diverse knowledge into urban practice. This education is essentially aesthetic and project-based, with little content in data management, public policy, sociology, and urban economics. Thus:

> "Obtaining information is difficult. Using the government's Active Transparency mechanisms to obtain the necessary data does not guarantee quality information, as much information is discontinued, georeferencing is precarious and uncoordinated between, for example, the Internal Revenue Service and the National Institute of Statistics. The distance between data and spatiality is problematic.

Non-urban information is terrible in Chile. The fragmentation of information in public entities is brutal; each entity has its data system" (I2).

"So, how can an architect influence the political sphere? Spatial practice is the particular project, but it is difficult to articulate people with space. That is the main objective of architecture as politics. How do architects see the land problem? There is an aesthetic exploration of the political relationship and architecture, but this is an accelerator of spatial practices, not of relevant political changes. On the other hand, there is no state policy on the public" (I21).

Likewise, the interviewees reflect little on the teaching role in critical training in the face of the city's urgent problems. Although, as mentioned earlier, there is a consensus on certain diagnoses, professional practice forces some positions to be changed to fit within the realm of the possible, of what will actually be funded and built. From the strong academic diagnosis, there is no real bridge to a pragmatic level of these diagnoses. Instead, there are adaptations in which the discourse does not necessarily match the work:

> "In academia, there is a certain coherent and consensual narrative, but even so, that discourse is not applicable. For example, Pablo Allard and his academic vision changed when working with Piñera; he had to do it. Academia does not include reality in its daily exercise. For example, profitability, there is a prejudice about talking about money in architecture, or at least in academia, resisting the logics of the capitalist model. Then, there is no deepening of the criticism of the capitalist model; the logics behind it are not understood" (I16).

Despite this situation, in recent years there have been instances where the common sense of the academic community has managed to generate urban agendas that defend the common good:

> "For example, for the new National Urban Development Policy, the deans had to form a block against the Chilean Construction Chamber. This, to install the need for a land policy. I am not exaggerating in saying that, for the first time in a long time, a group of architects is taking on a relevant role in the city, assuming significant ethical responsibility, placing the land in the city as something fundamental" (I18).

However, there are difficulties associated with the role of property in the development of urban practice: "there is trauma with agrarian reform and changes in land management. This is linked to private property and land" (I18). This "trauma" alluded to by the interviewee conditions what is or is not possible to carry out in practice, mainly because the right to property is extremely strong in Chile thanks to the way it is inscribed in the political constitution of the dictatorship of 1980. This ultimately determines an urban development model that focuses on some public elements such as infrastructure or norms, but always in line with economic powers:

> "When working on public policies, one must recognize the scope of what is possible, which is somewhat frustrating because radical positions are not implemented. Our lobbying capacity is vital in these processes. We work with the Chilean Construction Chamber and with the Association of Architecture Offices, but we also work with The Movement for a Just Reconstruction and with Ciudad Viva as well, with professional associations and universities" (I9).

> "The State is a partner and perhaps the main actor in urban economic power. It is a transmitter of wealth. From the State, the neoliberal discourse is invisible, but it is in practice. We believe that the State is fair, but in practice not so much. The shock doctrine normalized capitalism, and we still don't know how to get out of this mold" (I25).

The interviewees, although critical of the neoliberal model, rarely criticize the private world. In general, the views focus on the role of the State, which is tied hands in most urban aspects, precisely because of a pro-business regulatory framework. However, many

specifically critical views of market agents were not collected, who are often presented as a result of the lack of regulation. That is, the problem of the ethics of city production is somewhat absent in these diagnoses. Continuing with this view of the state's role, in Chile, space design is rarely performed from the state, although it does occur with some infrastructure elements. The urban form provided by the state is mainly centred on large spatial elements that contribute to facilitating the circulation of capital:

> "Chilean urban planners have little awareness of urban infrastructure, starting from the subway to highways, in terms of building the urban landscape. In Santiago, this landscape is mainly configured through infrastructures, and that, in general, is decided by politicians. There is a positioning of the new urban planners in the ears of politicians, however, their impact is not yet fully seen in the cities. On the other hand, equipment has not been understood as urban projects, but as individual and isolated architectural projects. Then, the Ministry of Public Works, in charge of most of these infrastructures, does not have an urban design department, for example" (I21).

In summary, urban practice under the Chilean neoliberal context has an important component associated with the state's role in the standards it sets for city development. In this sense, it does not go much further than figures and cases on the ideological structure of that state and its spatial results. There is also no view of the operational practice of urbanism in the city or specific criticisms of urban results. It seems that the answers to a better city would be in the State and not in a more ethical and less mercantilist professional practice. Far from being a bias, the convergence of similar views looks more like a diagnosis of the problem of the disciplinary practice of urbanism in Chile.

### 3.3. Facing Utopia

For Henri Lefebvre (1961), one of the greatest disciplinary virtues of urbanism is to be a field that must constantly work with the imagination of a different society, with what he called concrete utopia. That is, critical reflections on the current reality that became feasible projects according to the economic, social, methodological, technological, and political scopes of the time. In the case of urbanism in Chile, according to the results of this research, it seems that there is a consensus on the idea that utopia had to be renounced in favour of capital. "There is no imagination, it's not convenient", says interviewee 10, alluding to the fact that the imaginative nature of urbanism is not only unprofitable but risky for capital. "Gran Santiago lost the ability to imagine itself. The Metropolitan Regulatory Plan of Santiago 100 is an example, where only the urban limit is thought of and not the imagination of the city. There is no utopia. We are trapped by an image of a machine" (I2). This urban limit's main role is to define new areas for real estate exploitation and the production of new high-cost infrastructures.

Such urbanistic simplicity disguised as a metropolitan transformation plan is part of one of the interviewees' observations. They refer to the fact that initiatives from the state seem to be pushed with profitability objectives, not necessarily seeking the common good. Regarding this lack of imagination, many interviewees pointed to political parties, entities supposedly organised to propose alternative future ideas to the community, which become political projects. In political parties, the city is absent:

> "Those who were called to think about utopia did not do it, and today Santiago's scale became unapproachable. Moreover, with the return to democracy, speculation increased, and massive housing production. In the end, there is no thought of Santiago's democratic city; a false image of development was generated. The most critical urban planners were stunned; they did not know how to act. That's where the image of Santiago developed by investors is consecrated" (I11).

> "Political parties did not see the city as something relevant because the illusion that everything was going very well-made people believe that the city was not an issue. Today I am struck by the absence of the city in this structural discussion

of changes to come. The city is a collective and community exercise, and we are in a country where the model made the State's action retract with a subsidiary bias, making it absent in urban matters. The 1980 constitution is ideal for an authoritarian, so it relies heavily on the leadership of the nation's head and its priorities. But there is also the always important issue of private property and the power of private property consolidated in this Constitution" (I12).

Precisely, Chile faces an essentially imaginative process, such as drafting a new constitution for 2022. Many agendas are underway to propose some city ideas, however, much of those ideas have more to do with a memory than with new ideas. "The prospect of a future Santiago is explored in the past, 50 years ago, but not much today" (I9). It is frustrating to observe that an essential part of the utopian thinking of the interviewed urban planners refers to a past event, nostalgia for a political project truncated by the dictatorship. It can be argued that this truncated project is still desirable, but it can also be criticized that there are no other alternatives that do not necessarily refer to nostalgia as an imaginative alternative, which is actually an exercise in historical memory. On the other hand, in the face of the crushing neoliberal model, other views are more critical of the future:

"The extremist Chilean neoliberalism makes this city necessary for Latin America, both as an extreme business centre and as a reserve for capital, almost like a city-state. Cultural importation makes Santiago a kind of global neoliberal reserve. ...
I believe that in the future there will be a perfection of this market system" (I21).

"Based on Santiago as it is worked on in the urban today, I see an attractive product. I imagine a terrible copy of New York, that's how I imagine it, as we are today; and I'm optimistic." (I14).

From the interviews, it was challenging to obtain more imaginative descriptions; on the contrary, many of the comments on the idea of thinking about Santiago in the future were rather pessimistic. The neoliberal model is strongly criticized, and the frustration does not seem to come from the shortcomings of this model, but from not knowing how to offer a viable alternative to it. Not even a post-neoliberal city was presented among 27 experts separately, certainly surprising. There are some ideas of importing other urban images, but no alternatives that break with what would be a deepening of the current urban form. While there is no post-neoliberal urban image, there is more clarity about some urgent paths to reverse the perfection of the neoliberal urban utopia:

"Either we make a land reform, or [we will be much worse]. If we want to change Santiago, the State must act strongly in this. Politics is no longer able to manage indignation, the future is now in dispute, and here you can see where this city is going, towards the crisis" (I18).

Even so, the most imaginative solutions and ideas tend to refer to public policies and do not postulate new spatialities that push such visions. That is, there is no target image, but there is a diagnosis of the aspects of urban economy and the regulatory framework that keep the neoliberal city in a deepening process. This deficiency may be part of the absence of a political project that accompanies the urban image. That is, if a group of expert urban planners critical of neoliberalism cannot imagine a post-neoliberal city, it is difficult for any other social agent to do so.

In examining the 2019 social outbreak in Chile, a critical analysis of urban planning and political imagination reveals significant insights. The text underscores a departure from Henri Lefebvre's concept of 'concrete utopia' in urbanism, highlighting a shift in Chilean urban planning towards prioritising capital and profitability over imaginative, societal-centric urban spaces. This shift is emblematic of a broader political inertia, particularly within political parties and state initiatives, where urban development is driven more by market forces than by communal aspirations. The predominance of a neoliberal framework in Santiago's urban development is critically viewed as favouring elite interests and market mechanisms, neglecting the collective social vision. This approach has led to urban spaces that are more reflective of economic imperatives than of inclusive societal needs. The

interviews with urban planners reveal a palpable frustration stemming from the absence of viable alternatives to this neoliberal model, a sentiment that significantly contributed to the social unrest. The process of drafting a new constitution is seen as a pivotal opportunity for reimagining urban spaces, yet there is a noted tendency to look backward rather than forward, indicating a reluctance to propose innovative urban futures. This analysis suggests that the social outbreak can be partially attributed to a failure in urban planning and political imagination, where the inability to envision and create inclusive urban spaces resonated with the broader societal discontent, fuelling the unrest. The neoliberal approach, with its emphasis on market forces and private interests, emerges as a central factor in this urban planning failure, underscoring the need for a more imaginative and inclusive approach to urban development in Chile.

## 4. Discussion

Drawing from the methodological contributions of Bourdieu, to construct the habitus of Chilean urban planners, interviews have been conducted to recognize the conditions present in the agents, by identifying some cognitive and motivational structures in relation to the history and relationships of how these agents incorporated certain ways of operating that ultimately integrate and become spontaneous dispositions of the agents in each field (Bourdieu 1977, 1990, 1991). With the evidence obtained from the interviews, three typologies are determined (Figure 4) that describe the urbanistic subjectivity of the agents who have participated in this research.

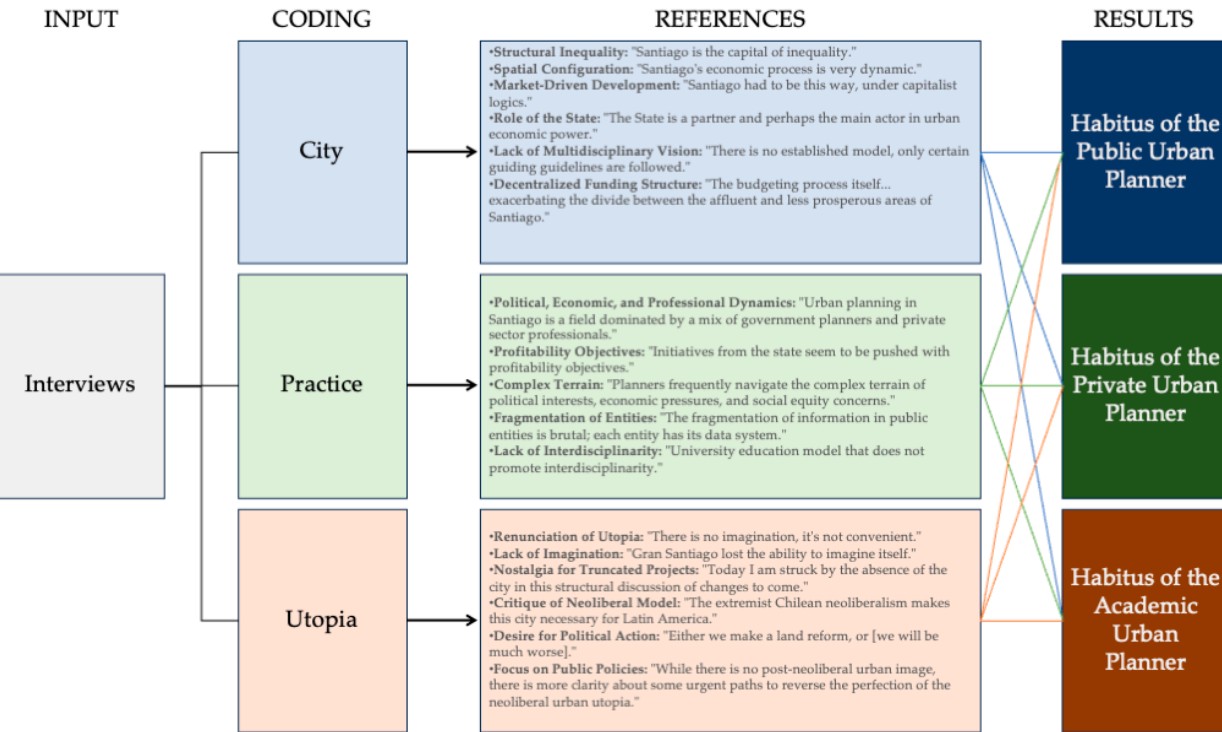

**Figure 4.** Diagram of discursive categorisation with the codes of each category.

### 4.1. Habitus of the Public Urban Planner

Characterised by working in state entities or to develop activities that directly depend on state funding from civil organisations. Their practice focuses on working with and understanding public instruments, developing non-profit activities, and feeling that their work is necessary for the development of the common good as one of the priorities of their regular tasks. This habitus is critical of the market that has shaped the city but recognises significant limitations in the regulatory field to regulate these urban processes. They also see the fragmentation of public entities as a disadvantage for the common good and an

advantage for market agents to organise and coordinate public priorities in accordance with political power.

This habitus focuses its diagnosis on an essentially unfair economic model. It sees public policy as an urban project, even when that policy does not have an urban form, which creates a dissociation between idea and result. It is challenging for this habitus to imagine urban forms, as its imagination applies to normative aspects or more systemic models than specifically spatial ones. The origin of this habitus is neoliberal technocracy itself, which constitutes a difficulty when developing initiatives outside that ideological framework, beyond being aware of the problems it generates.

### 4.2. Habitus of the Private Urban Planner

Characterised by developing their professional practice in the private sector, preferably with private financing, although they occasionally also work with public funds. Their practice focuses on developing commissioned works that usually adapt to the financial objectives of clients or their economic possibilities. In both cases, there are structural limitations to the practice related to economic reasons rather than the design and innovation capabilities of the habitus. Aware of these limitations, they see the neoliberal model as an original problem, which is the economic ordering principle of decisions on city matters as a blind spot. This is because the imperative to increase capital is predominant, and without a change in that logic, structural changes are difficult.

They call for a more present state in city production, although more from the perspective of financing initiatives that otherwise must be financed by private entities, as in the case of housing. It is a more imaginative habitus in space but with a pragmatic view of what is or is not applicable to reality. The origin of this habitus is the disempowerment of urban planners in city decision-making against the empowerment of financial capital to define where the most profitable space is developed, something that emerged more strongly in the mid-1980s.

### 4.3. Habitus of the Academic Urban Planner

Characterised by working in higher education institutions, teaching and researching urban matters, with institutional funds or public funds, which gives them somewhat more creative independence, but rarely do their ideas end in material works or normative projects. They have broader views on urban problems but lack an appreciation of the sense of what is achievable, that is, they tend to value the diagnosis more than the development of feasible proposals under prevailing conditions or that manage to transform those conditions. The academic process of urbanistic thought has information that crosses the habitus of the public and private urban planner, but it does not manage to shed the neoliberal framework of imagination. It is a habitus that also, when proposing, tends towards pessimistic or positions framed within the current neoliberal boundaries.

Despite operating in spaces protected by the research methodology itself, there are not many imaginative, groundbreaking, or utopian contributions that emerge from the findings. That is, the habitus of the academic urban planner knows how to read urban problems well, but something stops their imaginative processes, possibly the distance from spatial design or the absence of a professional need to achieve material results.

### 4.4. Theoretical Contribution

The literature, rooted in Bourdieu's conceptualization of habitus, provides a robust framework for understanding the dispositions and practices of professionals within their respective fields. In the context of Chilean urban planners, the findings of this study offer a nuanced and detailed exploration of how these professionals navigate the neoliberal urban landscape, revealing three distinct habitus: the public, private, and academic urban planner.

The habitus of the public urban planner, as showed by the findings, resonates with the literature's portrayal of professionals operating within state entities or activities reliant on state funding. Their commitment to the common good and their critical stance towards the

market-driven shaping of the city align with Bourdieu's notion of habitus as a system of dispositions shaped by historical and social contexts. However, their struggle to envision urban forms beyond normative or systemic models underscores the challenges they face in transcending the neoliberal technocratic origins of their habitus.

Similarly, the habitus of the private urban planner, characterized by their engagement in the private sector and their pragmatic approach to urban development, reflects the literature's emphasis on the economic imperatives that dominate the neoliberal city. Their call for greater state involvement in city production and their recognition of the structural limitations imposed by economic considerations highlight the tensions inherent in their professional practice. The findings suggest that while they are acutely aware of the neoliberal constraints, they grapple with the challenge of reconciling these with their professional and ethical commitments.

The academic urban planner's habitus, presents a unique juxtaposition of creative independence and a certain inertia in translating ideas into tangible urban developments. While they possess a broader perspective on urban challenges, their tendency towards pessimism or continuity, and their struggle to break free from the neoliberal imagination, align with Bourdieu's understanding of habitus as both a product and producer of social structures. The findings underscore the need for a more imaginative and transformative approach to urbanism, one that goes beyond diagnosing problems to envisioning and implementing alternative urban futures.

These findings offer a nuanced understanding of how the commodification of urban practices, rooted in neoliberal ideologies, deeply permeates the professional dispositions and actions of urban planners in Chile. By delineating three distinct habitus—public, private, and academic—the study illuminates the varying degrees to which market-driven imperatives influence urban planning decisions and strategies. While each habitus embodies unique challenges and perspectives, they collectively underscore the pervasive influence of neoliberal commodification on urbanism. This highlights the tension between market-driven objectives and the ethical and social responsibilities of urban planners, revealing the complexities and contradictions of navigating urban development within a commodified landscape.

The interplay between "the city, practice, and utopia" and the three identified habitus of Chilean urban planners—public, private, and academic—offers a rich tapestry for understanding urban development within a neoliberal context. The city, as a physical and social construct, is the canvas upon which these habitus enact their practices and ideals. For the public urban planner, the city is a realm of policy and public good, yet their practice is often constrained by the limitations of a neoliberal framework, hindering the realization of utopian visions that transcend market-driven imperatives. Their focus on normative and systemic models, rather than spatial innovation, reflects a habitus shaped by the historical and socio-political milieu, emphasizing the role of the state yet struggling against the grain of market forces.

In contrast, the private urban planner's habitus is deeply embedded in the neoliberal commodification of urban spaces. Their practice, primarily driven by market dynamics and private capital, shapes the city in ways that often prioritise economic imperatives over social or ethical considerations. This habitus, while pragmatic and spatially imaginative, is limited by the overarching neoliberal logic, constraining their ability to conceptualise and implement urban forms that diverge from profit-driven models. The tension between their professional expertise and the market realities underscores a conflicted relationship with utopian ideals, where visions of alternative urban futures are tempered by the prevailing economic order.

The academic urban planner, operating within the realms of theory and education, embodies a habitus that is both critical and constrained. Their broader understanding of urban issues allows for a more comprehensive critique of neoliberal urbanism, yet their practice often remains detached from tangible urban development. This detachment results in a habitus that is knowledgeable yet pessimistic, capable of diagnosing urban problems

but less effective in proposing feasible, transformative solutions. The academic urban planner's struggle to transcend the neoliberal imagination highlights a crucial gap between theoretical understanding and practical application, underscoring the challenge of bridging idealistic urban visions with the realities of contemporary urban practice. This dichotomy between theory and practice in the academic habitus reveals the complexities of actualizing utopian ideals within the constraints of the neoliberal city.

## 5. Conclusions

The neoliberal model has such an imposing force that in areas like urban planning it has managed to shape the habitus of urban planners, so that their decisions and operations align or do not really interfere with the rent-seeking objectives that the model demands of the city. This generates subjective positions that, while recognising a problem in neoliberalism, cannot break out of that field to explore proposals far from the margins established by the model itself. There is, then, an awareness of neoliberal action, but no detachment. Therefore, the habitus in its three interpretations presented here operate as a stance resulting from pragmatism, Chile's recent social history, and the ways to operate successfully in the neoliberal urban field, that is, in favour of capital profitability, whether from the public or private sector.

Unexpectedly, the academic urban planner's habitus seems unable to offer alternatives different from what exists; while it is critical of this world, its imagination does not go much further than what already exists in other neoliberal nations or from improving some existing ways of city-making in Chile. That is, based on the identified habitus, the absence of imagination is a common critical factor. In the current scenario, where a democratically elected collegiate body rethinks the political constitution of the republic, based on these urban planners, it can be said that there is not yet a target image of the post-neoliberal society on which to test new sections for a social agreement that aims, from urban disciplines in Chile, to overcome neoliberalism. This is seen as a pending and urgent task. The recent pandemic also brings urgent urban challenges, seeking healthier, less extended cities with better organizational capacity for a crisis.

Regarding to this work, in a revision to be developed on the ongoing changes in Chile, it will be essential and unavoidable to incorporate gender issues not only in the sampling process but also in the previous theoretical discussion. One of the most mentioned elements during the constituent discussion is to move towards a care city, something that has not appeared during the interviews analysed here but is a key factor within the framework of feminist urbanism (Moser 1993; Muxi et al. 2011). The sample can also be expanded to other regions of Chile, seeking more diverse approaches that allow producing more complex habitus typologies. This work, despite its limitations, presents a methodology that allows developing new methodological interpretations of the habitus applicable to spatial disciplines such as urban planning, architecture, or design. A deeper exploration of each of the habitus described here can also generate new specific research lines for each field of action that will be enriched by integrating factors associated with gender studies and perspectives from regions.

Considering the findings presented in this article, it's evident that the commodification of the city under the neoliberal model has not only reshaped the physical and social landscapes but has also deeply influenced the habitus of urban planners. The neoliberal paradigm, with its emphasis on market-driven solutions, has entrenched itself so deeply that even those critical of its implications find themselves constrained by its boundaries. This commodification has transformed cities from spaces of collective living and shared experiences into assets and investment opportunities. The urban planners, whether operating in public, private, or academic spheres, are navigating a terrain where the city's value is often reduced to its economic potential rather than its socio-cultural significance. This article underscores the urgent need for a reimagined urbanism, one that transcends the neoliberal confines and repositions the city as a space for people, not just profit. As cities continue to evolve, it is imperative that urban planning reclaims its visionary po-

tential, prioritising human-centric, sustainable, and inclusive urban futures over mere market-driven objectives.

**Author Contributions:** Conceptualization, F.V.-P. and M.A.-L.; methodology, F.V.-P. and M.A.-L.; software, F.V.-P.; validation, F.V.-P. and M.A.-L.; formal analysis, F.V.-P. and M.A.-L.; investigation, F.V.-P.; resources, F.V.-P. and M.A.-L.; data curation, F.V.-P. and M.A.-L.; writing—original draft preparation, F.V.-P. and M.A.-L.; writing—review and editing, F.V.-P. and M.A.-L.; visualization, F.V.-P.; funding acquisition, F.V.-P. All authors have read and agreed to the published version of the manuscript.

**Funding:** This research was funded by ANID Grant number 72140060. The APC was funded by Universidad de Las Américas Grant number 176.

**Institutional Review Board Statement:** The ethical approval for this research is Project ID 6168/001: Interpreting "The Urban Revolution": Segregative production of space in Santiago through a comparative study, granted by Chair of the UCL Research Ethics Committee (REC) of the University College London, UK in 2014. The study was conducted in accordance with the Declaration of Helsinki, and approved by the Institutional Review Board of UCL as indicated above.

**Informed Consent Statement:** Informed consent was obtained from all subjects involved in the study.

**Data Availability Statement:** Data available on request due to ethical restrictions and privacy agreements with interviewees.

**Conflicts of Interest:** The authors declare no conflict of interest.

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
