# Peer review of "Commodification in Urban Planning: Exploring the Habitus of Practitioners in a Neoliberal Context"

_socsci, doi:10.3390/socsci13010022_

Round 1
Reviewer 1 Report
Comments and Suggestions for Authors
This is a very good paper, exploring the model of urban planning in Santiago de Chile related to commodification of urban space and the predominance of regulatory mechanisms based on market profitability. The paper is well written, well organized and the flow of arguments is easy to follow. I have a few recommendations for making the paper analytically stronger.
I noticed that while the “social outbreak” started in October 2019, the 27 semi-structured interviews were conducted between January 2015 and August 2016 (page 2). Therefore, they were conducted before the period, where there seems to be some more awareness that regulatory market principles need to be more socially balanced. I think that this affects particularly section 3.3. Facing Utopia. The author(s) need to acknowledge that the interviews were conducted before this latest phase in urban planning, where a new constitution is being discussed. There are limitations to any research and the authors need to directly address this issue.
The three topics, as the authors refer to them, “the city, practice and utopia” (p. 6), are underdeveloped. There isn’t much of an explanation of how the authors have theorized these topics and how they are related to the main concept, habitus, introduced by Pierre Bourdieu. I think that the authors need to provide a better justification of those three topics and how they are situated within Bourdieu’s conceptualization of habitus.
In section 3.2. Field of Urban Practice, there is an interesting comment made by one of the interviewees (p. 12): “Chilean urban planners have little awareness of urban infrastructure, starting from the subway to highways, in terms of building the urban landscape. In Santiago, this landscape is mainly configured through infrastructures, and that, in general, is decided by politicians.” The authors further argue that the state plays an important role in setting the standards for city development.
I would like to see this aspect developed in much further analytical detail: How is infrastructure decided by the politicians? How is it possible for urban planners not to have awareness about infrastructure? What is the larger role and the influence of politicians in the sphere of urban practice? There are different types of politicians, as well, mayors, municipal authorities, legislatures, etc. The authors need to illuminate better the different types of interests that are intertwined in city development.
In this context, another interesting comment is mentioned, where different communes are fiscally independent (p. 9). This adds another layer in understanding the patterns of urban planning and how this kind of administrative fragmentation affects planning practices. This layer, however, is not explained at all in the paper and more attention needs to be given to it, particularly since it contributes to spatial inequality.
Finally, what is the link between the three topics, “the city, practice and utopia,” and the three types of urban agents, listed in section 4. Discussion. I think that overall, the authors need to strengthen the analytical aspect of their paper, explain better their main concepts, and establish stronger links between them. Overall, very good work!
Comments on the Quality of English LanguageThere are some minor grammar errors and sentences that can be improved.
Author Response
RESPONSES TO REVIEWER 1
GENERAL COMMENT: This is a very good paper, exploring the model of urban planning in Santiago de Chile related to commodification of urban space and the predominance of regulatory mechanisms based on market profitability. The paper is well written, well organized and the flow of arguments is easy to follow. I have a few recommendations for making the paper analytically stronger.
OBSERVATION 1: I noticed that while the “social outbreak” started in October 2019, the 27 semi-structured interviews were conducted between January 2015 and August 2016 (page 2). Therefore, they were conducted before the period, where there seems to be some more awareness that regulatory market principles need to be more socially balanced. I think that this affects particularly section 3.3. Facing Utopia. The author(s) need to acknowledge that the interviews were conducted before this latest phase in urban planning, where a new constitution is being discussed. There are limitations to any research and the authors need to directly address this issue.
RESPONSE 1: Thank you for your insightful observation regarding the timing of the interviews in relation to the "social outbreak" that began in October 2019. Your point about the evolving awareness of the need for socially balanced regulatory market principles, especially in the context of the ongoing discussions about a new constitution, is well-taken. To address this, we included a pargraph which explicitly recognizes this limitation. In this sense, the article aims to offer insights to explain the potential urban reasons of the social outbreak.
In Section 3.3 "Facing Utopia," we added an interpretation of how our findings are connected with the temporal scope of the social outbreak. We believe that this amendment will enhance the paper's contextual accuracy and provide readers with a clearer understanding of the scope and relevance of our findings within the rapidly evolving landscape of Chilean urban planning.
OBERVATION 2: The three topics, as the authors refer to them, “the city, practice and utopia” (p. 6), are underdeveloped. There isn’t much of an explanation of how the authors have theorized these topics and how they are related to the main concept, habitus, introduced by Pierre Bourdieu. I think that the authors need to provide a better justification of those three topics and how they are situated within Bourdieu’s conceptualization of habitus.
RESPONSE 2: Thank you for your valuable feedback on the development and theoretical framing of the three key topics: "the city, practice, and utopia," within the context of Pierre Bourdieu's concept of habitus. We appreciate your insight into the necessity of a more robust justification and theoretical linkage of these topics to habitus. We elaborate a whole new section to amend this flaw. By providing a more detailed justification of these three topics and their connection to Bourdieu's habitus, we aim to enhance the coherence of our argument and strengthen the theoretical foundation of our paper. This revision will not only address your concerns but also enrich the overall analysis and contribution of our study to the field of urban planning.
OBSERVATION 3: In section 3.2. Field of Urban Practice, there is an interesting comment made by one of the interviewees (p. 12): “Chilean urban planners have little awareness of urban infrastructure, starting from the subway to highways, in terms of building the urban landscape. In Santiago, this landscape is mainly configured through infrastructures, and that, in general, is decided by politicians.” The authors further argue that the state plays an important role in setting the standards for city development.
I would like to see this aspect developed in much further analytical detail: How is infrastructure decided by the politicians? How is it possible for urban planners not to have awareness about infrastructure? What is the larger role and the influence of politicians in the sphere of urban practice? There are different types of politicians, as well, mayors, municipal authorities, legislatures, etc. The authors need to illuminate better the different types of interests that are intertwined in city development.
RESPONSE 3: Thank you for highlighting the need for a deeper analytical exploration of the role of politicians in urban infrastructure decisions, the awareness of urban planners in this realm, and the varied political interests in city development. We acknowledge the significance of your points and propose to expand Section 3.2 "Field of Urban Practice" to address these aspects comprehensively. Specifically, we contextualize within the Chilean urban landscape, tackling its specificities, particularly in Santiago, to provide a grounded understanding of these dynamics. In this new version we briefly presented how it works urban planning in Chile and what is the role of politicians. By expanding on this aspect, we aimed to provide a richer and more comprehensive analysis that illuminates the complex interplay between political decision-making, urban planning practices, and infrastructure development. We believe that these enhancements will significantly contribute to a deeper understanding of the multifaceted nature of urban development in Chile.
OBSERVATION 4: In this context, another interesting comment is mentioned, where different communes are fiscally independent (p. 9). This adds another layer in understanding the patterns of urban planning and how this kind of administrative fragmentation affects planning practices. This layer, however, is not explained at all in the paper and more attention needs to be given to it, particularly since it contributes to spatial inequality.
RESPONSE 4: Thank you for drawing attention to the significant aspect of fiscal independence of communes and its impact on urban planning and spatial inequality. We agree that this represents a critical layer in understanding the patterns of urban development in Chile, particularly in terms of administrative fragmentation and its consequences. In light of your feedback, we include a new paragraph in the paper that specifically presents better the case.
OBSERVATION 5: Finally, what is the link between the three topics, “the city, practice and utopia,” and the three types of urban agents, listed in section 4. Discussion. I think that overall, the authors need to strengthen the analytical aspect of their paper, explain better their main concepts, and establish stronger links between them. Overall, very good work!
RESPONSE 5: Thank you for highlighting the need for a more explicit connection between the key themes of our paper and the typologies of habitus. We changed the discussion, adding new reflextions in order to strengthen our analysis of findings and connecting better with the new sections incorporated in the manuscript.
Reviewer 2 Report
Comments and Suggestions for Authors
1. The neoliberal perspective of this paper is relatively novel, which is more suitable for the urban development of Santiago, Chile, so the paper has obvious originality.
2. The clarity and relevance of the full text structure need to be enhanced, and there are many repeated statements before and after. If a theoretical frame diagram can be drawn, the logical structure of this paper will be significantly improved.
3. Section 1.1 of the paper, which should be revised.
4. The literature review of this paper is weak and needs further revision. Only in this way can the academic foundation, necessity and innovation of this research be effectively explained.
Author Response
RESPONSES TO REVIEWER 2
GENERAL COMMENT: The neoliberal perspective of this paper is relatively novel, which is more suitable for the urban development of Santiago, Chile, so the paper has obvious originality.
OBSERVATION 1: The clarity and relevance of the full text structure need to be enhanced, and there are many repeated statements before and after. If a theoretical frame diagram can be drawn, the logical structure of this paper will be significantly improved.
RESPONSE 1: Thank you for your valuable feedback on improving the structure and clarity of our paper. We recognize the importance of a coherent and logically structured presentation of our arguments and findings. Your suggestion to include a theoretical frame diagram is particularly appreciated as a means to visually represent the logical structure of our study. The diagram was now incorporated.
OBSERVATION 2: Section 1.1 of the paper, which should be revised.
RESPONSE 2: Thank you for pointing out the need for revisions in Section 1.1 of our paper. We are committed to ensuring that each section of our work contributes effectively to the overall narrative and scholarly rigor of the study. So, we ensured that the objectives of our study are stated clearly and concisely. This will involve revisiting our research questions and aims to ensure they are directly and effectively communicated in this section. We believe these revisions will strengthen the foundation of our paper, setting a solid groundwork for the subsequent sections and the overall argument..
OBSERVATION 3: The literature review of this paper is weak and needs further revision. Only in this way can the academic foundation, necessity and innovation of this research be effectively explained.
RESPONSE 3: Thanks for this observation. We incorporated 15 new references updating the content to make it more related to recent studies.
Reviewer 3 Report
Comments and Suggestions for Authors
The paper presented deals with an interesting and topical subject, but its design is imprecise and unclear. It is undoubtedly a worthy research that can be published, although it needs to make some adjustments to improve, as I say, its expository clarity.
The introduction discusses the current context in which capitalism, and therefore neoliberalism, seems to be, as Mark Fisher pointed out at the time, the only possible alternative. The authors start from this reality as a premise, but I do not see clearly the relationship between neoliberalism and urban planning. I think it would be good to make some modification in the introduction to try to clarify this aspect. Also, in line 54 there is an error in the quotation. On the other hand, I also fail to see clearly the relationship between Bourdieu's habitus and the research topic. In this sense, I suggest that you review the work of David Harvey to support the theoretical foundations of the paper.
The methodology used is clear and well developed although it presents two small problems. The first is related to the dates on which the interviews were conducted, and this should be pointed out. The second is related to the informants. There is a clear geographical bias that should be clarified and justified.
The results need some changes. In this sense, I think it would be convenient to include a graph with the main discursive categories and, if possible, with the codes of each category.
The discussion is possibly the section that needs the most changes. In this sense, the authors do not make a real discussion, but rather put the results obtained in dialogue with Bourdieu's work. I think it is necessary for researchers to look for other works on cities (Latin American, European, Asian, etc.) in order to try to contrast the information obtained in a more solid way. In the journal Urban Sciences, for example, there are several works that can be useful in this regard.
I think it would also be appropriate to include a section on limitations, indicating precisely the limitations of the article.
The conclusions are adequate.
As I indicated in the discussion, the bibliographical references should be significantly expanded.
I would like to encourage the authors to make the effort, since, as I said before, it is a meritorious article that could be published after the suggested changes.
Author Response
RESPONSES TO REVIEWER 3
GENERAL COMMENTS: The paper presented deals with an interesting and topical subject, but its design is imprecise and unclear. It is undoubtedly a worthy research that can be published, although it needs to make some adjustments to improve, as I say, its expository clarity.
OBSERVATION 1: The introduction discusses the current context in which capitalism, and therefore neoliberalism, seems to be, as Mark Fisher pointed out at the time, the only possible alternative. The authors start from this reality as a premise, but I do not see clearly the relationship between neoliberalism and urban planning. I think it would be good to make some modification in the introduction to try to clarify this aspect. Also, in line 54 there is an error in the quotation. On the other hand, I also fail to see clearly the relationship between Bourdieu's habitus and the research topic. In this sense, I suggest that you review the work of David Harvey to support the theoretical foundations of the paper.
RESPONSE 1: Thank you for your valuable feedback, particularly in pointing out the need for clearer articulation of the relationship between neoliberalism and urban planning, and the integration of Bourdieu's habitus concept into our research framework. We modified the introduction in order to provide a better articulation in this matter, incorporating references that are based on this review.
OBSERVATION 2: The methodology used is clear and well developed although it presents two small problems. The first is related to the dates on which the interviews were conducted, and this should be pointed out. The second is related to the informants. There is a clear geographical bias that should be clarified and justified.
RESPONSE 2: Thanks for this observation, we added more information to share the methodological strategy, temporality and geographical criteria of this research.
OBSERVATION 3: The results need some changes. In this sense, I think it would be convenient to include a graph with the main discursive categories and, if possible, with the codes of each category.
RESPONSE 3: Thanks, the graph was incorporated following your suggestion.
OBSERVATION 4: The discussion is possibly the section that needs the most changes. In this sense, the authors do not make a real discussion, but rather put the results obtained in dialogue with Bourdieu's work. I think it is necessary for researchers to look for other works on cities (Latin American, European, Asian, etc.) in order to try to contrast the information obtained in a more solid way. In the journal Urban Sciences, for example, there are several works that can be useful in this regard.
RESPONSE 4: Thanks for this observation. Certainly, while we incorporated new literature in different sections of the article, now we have better tools to discuss the findings based on Bourdieus approaches in Latin America, specifically.
OBSERVATION 5: I think it would also be appropriate to include a section on limitations, indicating precisely the limitations of the article.
RESPONSE 5: Thanks, the limitations of the article are now presented in the methodological section.
OBSERVATION 6: As I indicated in the discussion, the bibliographical references should be significantly expanded.
RESPONSE 6: The literature was expanded according to your suggestion.
Round 2
Reviewer 3 Report
Comments and Suggestions for Authors
The new version of the document has followed the suggestions correctly. The paper has been greatly improved. Some aspects that were previously not well defined have been clarified in this version and the justification of the decisions made by the investigators has been improved. New figures have been included that substantially improve the clarity of the process carried out and the results. Finally, the discussion has been improved by incorporating more information related to other research and other authors. Congratulations to the authors!